# Determination of the Content of Selected Pesticides in Surface Waters as a Marker of Environmental Pollution

**Ewa Pawłowicz-Sosnowska** [1], **Wioletta Żukiewicz-Sobczak** [2] [iD], **Paweł Sobczak** [3,*] [iD], **Maciej Domański** [2,4] and **Dominik Szwajgier** [5] [iD]

1. Faculty of Health and Social Sciences, Pope John II State School of Higher Education, 21-500 Biała Podlaska, Poland; e.pawlowicz@dydaktyka.pswbp.pl
2. Department of Food and Nutrition, Calisia University, 62-800 Kalisz, Poland; wiola.zukiewiczsobczak@gmail.com (W.Ż.-S.); maciej.domanski@interia.pl (M.D.)
3. Department of Food Engineering and Machines, University of Life Sciences in Lublin, 20-612 Lublin, Poland
4. Neuropsychiatric Hospital in Lublin, 20-442 Lublin, Poland
5. Department of Biotechnology, Microbiology and Human Nutrition, University of Life Sciences in Lublin, 20-704 Lublin, Poland; dominik.szwajgier@up.lublin.pl
* Correspondence: pawel.sobczak@up.lublin.pl

**Abstract:** Pesticides are a major problem not only in the food chain but also when considering the protection of our planet. The use of neonicotinoids has been banned in the EU due to their high toxicity to living organisms, in particular honeybees. The presence of neonicotinoids in natural waters poses a threat to pollinating insects and thus hampers organic production. Pesticide residues in the natural waters of agricultural land are monitored within the framework of promoting sustainable rural development to maintain the safety of human and animal health. Chromatographic analyses of selected neonicotinoid pesticides in water samples from agricultural sites in eastern Poland were performed using high performance liquid chromatography (HPLC), which followed a solid-phase extraction (SPE). The research revealed no evidence of neonicotinoids contamination. Water quality in this region can be a good factor in promoting sustainable development. The obtained results complement the existing knowledge on the impact of neonicotinoids on both the sustainable food chain and the environment. Based on the results obtained, it is possible to conclude that they are not being used in rural area under the study.

**Keywords:** water; health; neonicotinoids; sustainable development; pollution of natural waters; eastern Poland





## 1. Introduction

Neonicotinoids were developed in the late 1980s and introduced into the pesticide market in the 1990s. Imidacloprid is the oldest neonicotinoid and was first approved for use in the United States in 1994 and in Canada in 1995 to control damage to potatoes, tomatoes, apples, cornsalad and other greenhouse plants [1]. Unlike other neonicotinoids (i.e., acetamiprid, thiacloprid, thiamethoxam and imidacloprid), it was not approved for crops for human consumption in Europe until 2004 [2]. The use of neonicotinoids was registered in around 120 countries worldwide [3]. There are huge benefits associated with the use of insecticides in agriculture, public health, forestry and countryside landscapes, which greatly contribute to the development of the economies of developed and developing countries [4]. Extensive use of insecticides is, among many factors, partly responsible for the increase in yields in agriculture production in recent decades. For example, wheat production in the UK and corn production in the United States increased over the past three decades [4]. However, the use of neonicotinoids is associated with serious health consequences for people and the environment [4]. Over the last decade, there has been overwhelming evidence about the potentially harmful impact on humans, non-target

insects, aquatic invertebrates as well as side effects affecting the environment associated with the use of specific classes of insecticides [5,6].

Recent studies of surface water showed a worldwide presence, at various rates and concentrations, of several neonicotinoids, in particular imidacloprid, clothianidin and thiamethoxam [7]. The evaluation focused on the surface water in eight countries showed that the neonicotinoids were frequently detected in surface water at concentrations varying from 0.001 µg/L to 320 µg/L. These determinations prompted countries around the world to reduce the use of neonicotinoids in order to minimize the potential adverse effects on non-target organisms such as bees and macroinvertebrates [2].

Other researchers investigated the occurrence of neonicotinoids in aqueous environments and degradation mechanisms under laboratory conditions, for a global survey of the neonicotinoids' lifecycle, their transport mechanisms and their degradation under realistic environmental conditions is limited. A comprehensive overview of the neonicotinoids, their occurrence and interference with surface waters were presented.

The growth of neonicotinoids in the global market is backed by their effectiveness in minimizing crop damage. Nowadays, neonicotinoids dominate about one-quarter of the world market of insecticides, in which 80% of insecticides are used for seed treatment [8].

The three neonicotinoids most commonly used in urban and agricultural environments are imidacloprid, thiamethoxam and clothianidin [7]. In 2009, imidacloprid accounted for 41.5% of the purchased neonicotinoids with the market value of USD 2.6 billion, followed by clothianidin and thiamethoxam in the amount of USD 627 and 439 million, respectively [9]. However, since 2014, neonicotinoids were registered in more than 120 countries worldwide, of which thiamethoxam accounted for 25% of the total global market of insecticides with a market value of USD 3 billion, while in 2012, imidacloprid and clothianidin were the most used insecticides accounting for 85% of the global insecticides market [3].

Natural groundwater quality determines a number of important health and environmental aspects. One of them is the fact that certain compounds might, in varying degrees, be toxic to living organisms. Polycyclic aromatic hydrocarbons, polychlorinated biphenyls and dioxins are well known due to their impact on real environmental problems and have gained considerable publicity and research devoted to them [10,11]. Pesticide residues and products of their metabolism may also penetrate the water. In addition to their wide practical use in agriculture, horticulture, orchard cultivation and forestry, it is also important to understand the way in which they migrate and have an impact on the environment [12,13]. There are serious concerns about the potential toxic effect of pesticides on non-target organisms, including humans. The toxicity of pesticides is associated with the increased risk of exposure to neurological diseases and related to dysfunctions affecting humans [14]. Many pesticides containing neonicotinoids, whose industrial manufacture and used were banned in 2018 by the Regulation of the European Commission due to their high toxicity to living organisms, in particular honeybees. In July 2018, the Polish Ministry of Agriculture announced a derogation from the ban on the use of neonicotinoids in the case of two substances from the neonicotinoids group, viz., clothianidin and thiamethoxam, in rapeseed seed treatment. For some time, there has been a reduction in the number of bee colonies in Poland, while the pollination needs are increasingly bigger. Pesticide residues present in natural water constitute a threat of extinction of pollinating insects and thus a hindrance of natural pollination processes. This leads to negative consequences in crop production [1,9,15]. Therefore, it is justified to monitor the waters for the presence of selected pesticides and in particular neonicotinoids, especially in rural areas [2]. Due to the bioaccumulative nature of pollutants, new combinations of toxic compounds are formed, which may be even more harmful than individual pollutants. It should also be remembered that many living organisms, both invertebrates and vertebrates, are part of the human food chain [3].

For that reason, monitoring the quality of water present in the environment becomes extremely important. Obtaining such data is necessary for the protection of the environment

against adverse biological effects resulting from repeated chemical pollution caused by conscious or unconscious human existence [4].

At the same time, promoting sustainable development of the rural areas in Poland, including the use of so-called clean technologies, is of growing importance [5–7]. This approach takes into account the need to stimulate competitiveness in the economy, but with separating economic growth from environmental degradation, as well as a cross-sectoral approach to solving social problems [8]. The main idea behind the sustainable development concept is the preservation of the environment and natural resources for future generations also by changing the model of civilization development to one based on the consumption model creating lesser pressure on the environment and with a changed value system. Scientific research has an important role to play in this respect [16]. Almost half of the EU territory is used for agriculture. It was, therefore, recognized that the effects obtained in agricultural activities largely determine success on a global scale. Organic farms are farms using organic methods of agricultural production, with certificates issued by a certification body or those in the process of conversion to organic methods of agricultural production (under the control of a certification body). The guiding principle of the organic system is the cultivation of plants according to the standards of good agricultural practice and with due attention to the phytosanitary conditions of plants and soil protection. The main characteristics of organic farming are sustainable plant and animal production, a ban on the use of synthetic fertilizers, pesticides and fodder additives and the use of natural biological and mineral resources. This involves the use of animal manure, plant debris (compost), crop rotation, weed and pests control implemented by using biological and mechanical methods and the protection of animals and plants [17].

Organic farming is the most environmentally friendly type of agricultural production. The use of natural methods, without agrochemicals, leads to the preservation of biodiversity and the protection of natural resources. Food produced in such conditions is free from contamination and is characterized by high quality parameters. According to the Agricultural and Food Quality Inspection (IJHARS), as of 2017, Lublin province is fifth in Poland and third in Eastern Poland when it comes to the number of organic producers. More than 2000 certified farms in the region, 2061 to be precise, do not use chemical pesticides and fertilizers, fertilizing only with green manures and minerals available in nature, while in animal nutrition, they use predominantly their own organic fodder [18].

Water, being the substrate for all life processes, plays an important role in the biosphere and has a strong impact on the living conditions of all living organisms. That is why monitoring surface waters, which carry a huge amount of organic matter of anthropogenic origin, is so important [19,20].

The aim of the study is to perform a qualitative assessment of water for the presence of contamination with selected pesticides, i.e., neonicotinoids, as the indicator of their irresponsible use on agricultural farms. Chromatographic analyses of selected neonicotinoid pesticides present in water samples from agricultural sites in eastern Poland were performed using a high-performance liquid chromatography (HPLC), which followed a solid-phase extraction (SPE). It was assumed that excessive use of pesticides in agriculture, horticulture, orchard production, and forestry contribute to the contamination of natural waters and constitute risks for human and animal health and thus for organic production.

## 2. Materials and Methods

For the HPLC test, a set 25 of water sampling sites was designated within the Lublin region. From each location, samples were collected twice during the periods of pesticides application in orchards and cultivated fields, viz., in the timeframe between the fall of 2016 and spring of 2017. The designated locations are: Biała Podlaska, Bokinka Królewska, Bordziłówka. Dębowa Kłoda, Jedlanka, Kobylany, Kodeń, Kolonia Zakalinki, Kołczyn, Kożanówka, Łęgi, Łukowce, Małaszewicze, Międzyrzec Podlaski, Mokre, Neple, Niedrzwica Duża, Niedrzwica Kościelna, Ogrodniki, Piszczac, Rakowiska, Rogoźnica, Rossosz, Sławatycze and Terespol (Figure 1).

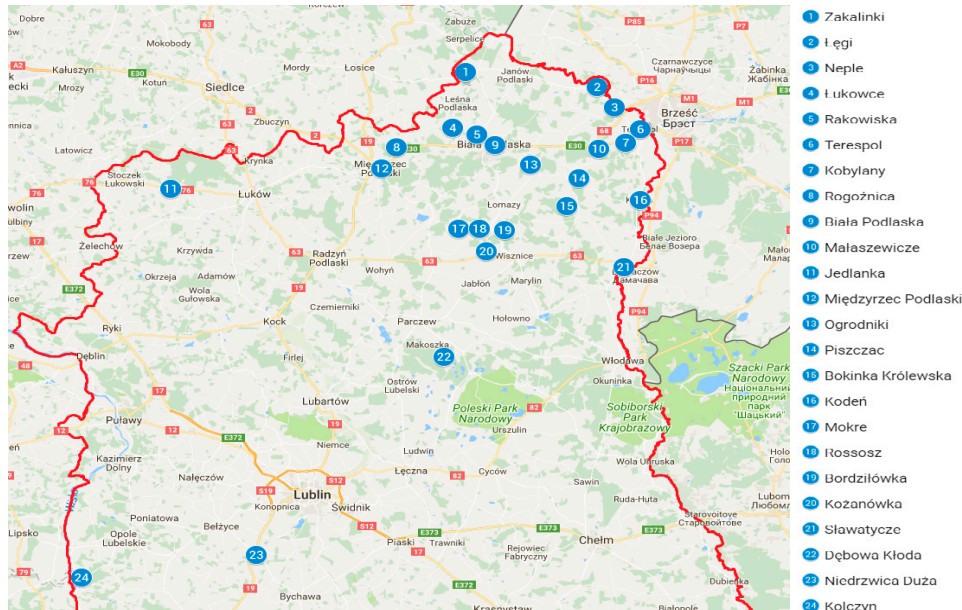

**Figure 1.** A fragment of a map of the Lublin voivodship with the location of the collected water samples—own research conducted in 2016–2017.

The water samples were collected in special plastic containers (bottles) in volumes of 5 litres from drainage ditches located along crop fields, especially of rapeseed and fruit orchards, in the vicinity of local rivers. Insecticides most commonly used since the early 1990s, i.e., neonicotinoids, were deliberately chosen for the analysis of the water samples from agricultural land. Despite their effectiveness in improving the protection and crop management, these agrochemicals were analysed due to their disturbing negative impact on non-target species, such as honeybees and aquatic invertebrates. In recent years, neonicotinoids have been detected in rivers and streams worldwide [21–23].

Neonicotinoids are insecticides that exhibit physicochemical properties, rendering them more useful over other classes of insecticides. This includes having a wide range of application techniques (e.g., foliar, seed treatment, soil drench and stem applications) and high efficacy in controlling sucking and biting insects (i.e., whiteflies, thrips, leaf miners, beetles and lepidopteran species) [24].

Pesticide content was determined using the HPLC method in accordance with the developed test procedure. A liquid chromatograph Dionex Thermo Scientific, equipped with a UV detector, was used for the analysis with a wavelength of 255 nm. For the separation, a Thermo Scientific reverse-phase column Aquasil C18 with the dimensions 250 mm $\times$ 4.6 mm 5 μm was used together with LC$-$18 column protection. A mobile phase of acetonitrile/water 30:70 (*v/v*) was used for the testing. Injection volume was equal to 75 mL; the flow rate was 1 mL/min. The column thermostat was set at 25 °C. Prior to chromatographic analysis, 1 L of water sample was extracted in two repetitions using an SPE C18-type cartridge (500 mg, 6 mL). Before the extraction, the columns were activated with 5 mL of methanol and 5 mL of distilled water. After the extraction, samples were eluted with 5 mL of methanol. After evaporation of the solvent, the samples were dissolved in 1 mL of mobile phase and analysed using a high-performance liquid chromatography (HPLC) with DAD UV-Vis detector. Each sample was tested twice. The research was carried out at EKO-AGRO-TECH at PSW Biała Podlaska.

## 3. Results

The analysis was carried out to determine the content of the selected 5 neonicotinoids, viz., acetamiprid, clothianidin, imidacloprid, thiacloprid and thiamethoxam, in the 25 examined samples of water from agricultural land in the Lublin province. It was con-

firmed through the interviews with 15 farmers and 10 fruit growers that next to 25 points where water from drainage ditches was sampled for testing, pesticides had been used in the spring period in the neighboring 12 rapeseed cultivations, 3 cereal crops cultivations and 10 fruit orchards:

- in rapeseed fields before the period of rapeseed flowering in May;
- in the cereal crop fields and orchards in April, May and June;
- and in orchards also in the summer.

Based on the community interview, it was found that the farmers recognised that good yields are obtained using crop protection products and that these products are used in accordance with the financial means—the larger the farm, the more products are used. Products containing neonicotinoids were applied as foliar sprays and as seed coatings in seed treatment.

Calibration curves within the range from 0.1 to 1 µg/L were prepared for quantitative determination. The limit of detection for all analysed neonicotinoids was 0.5 µg/L.

The wavelength for acetamiprid, clothianidin, imidacloprid, thiacloprid and thiamethoxam was set at 255 nm.

In the next step, the retention time of the standard of the five tested neonicotinoids was determined using high-performance liquid chromatography (HPLC) with a diode array UV-Vis detector. The retention time is as follows: acetamiprid—12.63 min, clothianidin—8.75 min, imidacloprid—10.20 min, thiacloprid—18.77 min and thiamethoxam—6.86 min.

The quantitative analysis did not reveal the presence of the 5 neonicotinoids in the samples of water for which permissible levels of pesticides are shown in Table 1.

**Table 1.** Permissible levels of pesticides in particular categories of water quality according to the Regulation of the Minister of Environment of 27 November 2002 [25].

| Category of Water Quality | Pesticides in Total | Water Quality |
|---|---|---|
| A 1 | up to 0.001 mg/L (1 µg/L) | water requiring a simple physical treatment |
| A 2 | up to 0.0025 mg/L (2.5 µg/L) | water requiring a typical physical and chemical treatment |
| A 3 | up to 0.005 mg/L (5 µg/L) | water requiring high-efficiency chemical and physical treatment |

The assumed test hypothesis regarding the presence of neonicotinoids in the tested water samples was not verified positively, which could result from a number of external factors, notably the occurrence of frequent rainfall in spring and the nearby presence of local rivers, which could affect the dilution of the products containing neonicotinoids below the detection levels.

Based on the literature query, we assumed that neonicotinoids are well soluble in water and have a long period of decay. They can, therefore, be present and active in the water, soil, crop cultivation, follow-up crops and plants accompanying crops, even long after they were originally used in a particular crop [26].

## 4. Discussion

The results of the water quality tests carried out in the Lublin region for the presence of contamination with selected neonicotinoid pesticides show no contamination with the residues of pesticides used in cultivated land and orchards within the examined rural areas. Therefore, monitoring water quality in rural areas can be a good factor in promoting the sustainable development of rural areas within the discussed region. Rich environmental resources characterising the Lublin region must be seen in terms of the "ecological capital" that is available to local communities. Taking care of the natural environment can also be recognized as an opportunity for development and not only as an obstacle, even though some contradictions between the natural environment and the economy are inevitable in

some cases. It is necessary to develop in the public awareness the perception of ecological values in terms of the public interest [27]. There is a significant demand for organic products in the EU states, Japan and the USA. Therefore, further investments in vegetable and fruit production would provide an opportunity for organic farming in Eastern Poland. Financial support for organic farms could increase the range of produced food. It is necessary to educate the public of the region further and to diversify distribution channels. National, regional, and local level authorities, as well as producers of organic products themselves, should all actively promote organic farming, for instance, on social media, as this type of production is beneficial for the entire economy. This should contribute to the increased production and sales of organic products both in the domestic and foreign market, and consequently, it should also strengthen the position of Poland as a leading producer and processor of organic food [28]. Organic farmers should be valued and encouraged to act as leaders in local sustainable development [18,29]. The obtained results are complementary to the existing knowledge on the impact of neonicotinoids on both the environment and the life of bees that are presented in many publications [30–37].

### 5. Conclusions

Based on the conducted research, the following conclusions have been formulated:

1. The study carried out did not reveal contamination with the selected pesticides (acetamiprid, clothianidin, imidacloprid, thiacloprid and thiamethoxam) in the tested water reservoirs. These results, however, do not provide reliable answers, and it is worth extending the tests to include samples of the pollen and nectar of bees, in which other studies revealed the presence of the above-mentioned neonicotinoids. The use of pesticides in crop protection should be controlled, also taking into account interviews with farmers and fruit growers as well as analyses of the external factors at environmental sampling points. In addition, the issue of the combined use of several pesticides is one of the important areas where more research is needed.
2. The quality of water in the region under study can be an important factor in promoting the sustainable development of the food chain.

**Author Contributions:** E.P.-S.—investigation, formal analysis; W.Ż.-S.—data curation, methodology; P.S.—writing—original draft preparation, conceptualization; M.D.—software, resources; D.S.—validation, visualization. All authors have read and agreed to the published version of the manuscript.

**Funding:** This research received no external funding.

**Data Availability Statement:** Not applicable.

**Conflicts of Interest:** The authors declare no conflict of interest.

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
