# Peer review of "Determination of the Content of Selected Pesticides in Surface Waters as a Marker of Environmental Pollution"

_sustainability, doi:10.3390/su13168942_

Round 1
Reviewer 1 Report
The authors report an interesting case study regarding a selected pesticide survey in the area of Lublin. While the motivation of this study is quite understandable, and the draft focuses a qualitative case study, more data what be needed to verify statements made within this draft such as „it is possible to conclude that they are not being used in rural area under the study“ This conclusion ist not supported as such regarding the low database and missing comparsion to other studies. Moreover just five out of a much greater spectrum of pesticides have been investigated here, and therefore a general statement like the one stated above is not acceptable.
Overall the following aspects of this manuscript would need a serious revision:
- Whole manuscript: A lot of the literature is > 5 years old / a modern state of the art ist missing, respectively regarding the Introduction/Discussion sections.
- What would be expected in the Result-section is at least an overview of the precise results for all 25 test-sites in 2016/2017 periods.
- As mentioned above, „selected pesticides“ not found at the 25 sampling points cannot not be a profound basis for the whole Lublin region, or even reflect the pesticide-contamination level in the region adequately. If such a „selected pesticide survey“ is done here no generalisation can be done in view of conclusions. This should be reformulated.
- The idea of organic farming is an important one, and reasonably right mentioned here. However the Discussion-section should also be used with regard to precise result comparisons of this study to previous literature studies, both on the „selected 5-pesticides“ presented here to elsewhere in Poland/Lublin/ national/international rural areas … same for the whole pesticide-spectrum in the rural national/international context.
- Whole manuscript: Some typo mistakes are still found in the text regarding word division e.g. abstract (fac-tor) /introduction (cultiva-tion, environ-ment, un-conscious) an so forth.
Author Response
After carefully reading the reviews, let us begin with the words of thanks for the very valuable and insightful analyzes and opinions of your honorable Reviewers, which inspire further efforts to correct this study. All valuable comments and tips motivate us to further, even more intensive research work.
Below, we provide substantive answers to the analyzes and recommendations of the esteemed reviewer.
The most commonly used insecticides since the early 1990s - the neonicotinoids - were deliberately selected for the analysis of agricultural water samples. Despite their effectiveness in improving crop conservation and management, these agrochemicals have been analyzed for their negative effects on non-target species such as honeybees and aquatic invertebrates. In recent years, neonicotinoids have been detected in rivers and streams around the world, write Borsuah J.F., Messer T.L., Snow D.D., Comfort S.D., Mittelstet A.R. in 2020 in Literature review: global neonicotinoid insecticide occurrence in Aquatic Environments. Water 12 (12): 3388 (the position of the literature is included in the references). Their purpose was to provide a comprehensive overview of neonicotinoids with an emphasis on their fate and transport mechanisms to and within surface waters and their distribution in waterways around the world. A better understanding of the mechanisms of fate and transport will enable scientists to: accurately predict the occurrence and persistence of insecticides entering surface waters and the potential for exposure to non-target organisms in regions with intensive agriculture. Neonicotinoids are insecticides that exhibit physicochemical properties that make them more useful over other classes of insecticides. This includes having a wide range of application techniques (e.g. foliar, seed dressing, soil drench, and stalk application) and being effective at controlling suction and biting insects (i.e. whitefly, thrips, leaf miners, beetles and butterflies).
More literature were added:
- Hrynko, I. Optimization of the methods for the determination of 7 neonicotinoids in honey bees, honeys, melliferous weeds and guttation fluids. Progress in Plant Protection 2021, (1), 82-92, DOI: 10.14199/ppp-2021-010.
- Borsuah, J.F., Messe,r T.L., Snow, D.D., Comfort, S.D., Mittelstet, A.R. Literature review: global neonicotinoid insecticide occurrence in Aquatic Environments. Water 2020, 12 (12), 3388. DOI: 10.3390/w12123388.
- Valverde, S., Ibáñez, M., Bernal, J.L., Nozal, M.J., Hernández, F., Bernal, J. Development and validation of ultra high performance-liquid chromatography-tandem mass spectrometry based methods for the determination of neonicotinoid insecticides in honey. Food Chemistry, 2018, 266: 215–222. DOI: 10.1016/j.foodchem.2018.06.004.
- Jeschke, P.; Nauen, R. Thiamethoxam: A neonicotinoid precursor converted to clothianidin in insects and plants. Acs Symp. Ser. 2007, 948, 51–65.
- Inspection of Environmental Protection, Voivodship Inspectorate of Environmental Protection in Lublin 2018. Report on the state of the environment of the Lubelskie Voivodeship in 2017, Environmental Monitoring Library, Lublin. http://www.wios.lublin.pl/srodowisko/raporty-o-stanie-srodowiska/
- Monitoring the implementation of the Sustainable Development Goals in Poland Information as at the end of 2020 Ministry of Development, Labor and Technology Warsaw, June 2021. file:///C:/Users/pracownik/Downloads/Monitorowanie_CZR_w_PL.pdf
Water samples were collected in special 5-liter plastic containers / bottles from drainage ditches along arable fields, especially rape and fruit orchards. near the surrounding rivers. The most commonly used insecticides since the early 1990s - the neonicotinoids - were deliberately selected for the analysis of agricultural water samples. Despite their effectiveness in improving crop conservation and management, these agrochemicals have been analyzed for their alarming negative effects on non-target species such as honeybees and aquatic invertebrates. In recent years, neonicotinoids have been detected in rivers and streams all over the world.
The Environmental Protection Inspectorate, the Provincial Inspectorate for Environmental Protection in Lublin conducts constant monitoring of the state of the environment in Poland and the reports also include the physical and chemical parameters of natural waters. therefore, we did not include it in the research results presented in this study.
The section “Discussion” were improved (text marked in the manuscript).
The “Results” was changed. The sentence was added: These results, however, do not provide reliable answers and it is worth extending the tests to include samples of pollen and nectar of bees, in which other studies revealed the presence of the above-mentioned neonicotinoids.
Reviewer 2 Report
The work is important from the point of view of the availability of information on environmental pollution by hazardous pesticides in a specific region of Poland. It informs about the culture of organic farming in the study area.
It would be useful to add to the keywords: pollution of natural waters, eastern Poland.
Unclear: What were the 25 water samples? How were they taken and from what water bodies: rivers, streams, lakes, puddles after rain, irrigation water, soil solution, etc. ???
Author Response
After carefully reading the reviews, let us begin with the words of thanks for the very valuable and insightful analyzes and opinions of your honorable Reviewers, which inspire further efforts to correct this article. All valuable comments and tips motivate us to further, even more intensive research work.
Below, we provide substantive answers to the analyzes and recommendations of the esteemed reviewer.
The keywords was reached by added: pollution of natural waters, eastern Poland
The water samples were collected in special plastic containers (bottles) with the volume of 5 litres, from drainage ditches located along crop fields, especially of rapeseed and fruit orchards, in the vicinity of local rivers.
Reviewer 3 Report
This paper studies the important question of environmental contamination with nionicotinoids through an environmental sampling program. The study found no contamination in any samples. Drawing conclusions from a negative study requires a careful, thoughtful research design. Unfortunately, they are more difficult to do well than a positive study.
There are several significant limitations to the study.
- There is no description or data related to the water samples. What was the source of the water? Streams, ponds, irrigation wells, drinking water supplies….? How close were they to the farmers fields?
- There is little information on the type of agriculture in the area. We learn that there are 2,016 organic farms in the region, but what is the denominator? What types of crops are being grown? Are these crops on which nionicotinoids are currently being used?
- There is no data to indicate that the water would have been contaminated if nionicotinoids were in use. For example, the authors could check for the presence of chemicals or evidence of fertilizers in use at the farms in the area, such as nitrates or coliform (from manure). Alternatively, they could have taken samples in an area known to be using these pesticides to show that their presence would have been detectable.
Figures 2 and 3 are not necessary. These are simply calibration curves.
Your conclusions are not supported by the data. Specifically:
- This is just a statement of fact, so not really a conclusion.
- There was no assessment of the importance of water quality in promoting a sustainable food chain.
- Without the types of evidence described in item 3 above, there is no proof that the failure to detect means the absence of use.
The absence of evidence is not, by itself, the evidence of absence.
Author Response
After carefully reading the reviews, let us begin with the words of thanks for the very valuable and insightful analyzes and opinions of your honorable Reviewers, which inspire further efforts to correct this article. All valuable comments and tips motivate us to further, even more intensive research work.
Below, we provide substantive answers to the analyzes and recommendations of the esteemed reviewer.
1 and 2. The water samples were collected in special plastic containers (bottles) with the volume of 5 litres, from drainage ditches located along crop fields, especially of rapeseed and fruit orchards, in the vicinity of local rivers. The crops were used during tests.
According to the Commercial Quality Inspection of Agricultural and Food Products, the number of organic producers in Poland as of December 31, 2020 is 20,274, and in the Lubelskie Voivodeship it is only 10.13% - 2,054 of all organic farms in Poland. In turn, the area of ecological agricultural land in Poland as at December 31, 2020 is 108439.01 ha, and in the Lubelskie Voivodeship it is 3.43% - 3778.15 ha. Due to the fact that organic producers cannot legally use pesticides, an attempt was made to check the presence of the neonicotinoids used in the protection of rapeseed and fruit production in selected farms in the Lubelskie Voivodeship.
- According to the Statistical Office in Lublin, the consumption of fertilizers in 2013 was presented in the Characteristics of agricultural holdings in the Lubelskie Voivodeship. Data on mineral fertilizers, calcium fertilizers and natural fertilizers of animal origin used in a farm cover the period from June 2, 2012 to June 1, 2013. The consumption of mineral (nitrogen, phosphorus, potassium) and calcium fertilizers is presented per pure NPK component and CaO. Conversion of fertilizer consumption per 1 ha of agricultural land in the 2012/2013 agricultural year was made to the area of agricultural land in good agricultural condition as of June 1, 2013. According to the 2013 results, approx. 154 thous. farms (86.2% of all farms) used mineral, calcium or natural fertilizers for the 2013 harvest. The share of farms using mineral, calcium and organic fertilization of animal origin in the total number of farms running agricultural activity was, respectively, 78.0%, 9.9% and 44.0%. Nitrogen fertilizers, as the basic yield factor, were used by 74.2% of all farms (67.9% in Poland). Due to the wider operation, users also often used compound fertilizers - 22.3% of farms (21.6% in Poland). Much fewer farmers enriched the soil with fertilizers containing phosphorus and potassium. Phosphorus and potassium fertilizers were used by 6.6% and 5.6%, respectively, (in Poland - 4.8% and 4.2%, respectively) of farms. Calcium fertilizers, extremely important due to the acidity status of Lublin soils, were introduced into the soil by 9.9% of farm users (6.8% in Poland)
Fig. 2 and Fig. 3 was deleted
The conclusions was changed by added: These results, however, do not provide reliable answers and it is worth extending the tests to include samples of pollen and nectar of bees, in which other studies revealed the presence of the above-mentioned neonicotinoids.
The conclusion 3 were deleted.
Round 2
Reviewer 1 Report
The authors have updated some literature sources in the materials and methods and discussion sections. The whole information now given in the discussion section is structurally at the wrong place and should be integrated in a proper way in the introduction section.
Partly some improvements are seen. However the single results of the 25 stations are still missing. Though some more literature is discussed ithe overall context to the results is missing. In terms of transparency the results of the single stations should be clearly mentioned, both in the result section as well as adequately in terms of comparisons to international scale in the discussion section.
Author Response
Response to the Reviewer 1
Thank you again for your suggestions to better improve our manuscript. We made necessary changes according to your remarks:
- We moved informations given previously in discussion section to the introduction chapter.
- We improved our results to adding information of our samples: It was confirmed through the interviews with 15 farmers and 10 fruit growers that next to 25 points where water from drainage ditches was sampled for testing, pesticides had been used in the spring period in the neighboring 12 rapeseed cultivations, 3 cereal crops cultivations, and 10 fruit orchards:- in rapeseed fields before the period of rapeseed flowering in May, - in the cereal crops fields and orchards in April, May, and June, - and in orchards additionally also in the summer. Basing on the community interview it was found that the farmers recognised that good yields are obtained using crop protection products and that these products are used in accordance with the financial means - the larger the farm, the more products are used. Products containing neonicotinoids were applied as foliar sprays and as seed coating in seed treatment.
And adding information of international results:
The assumed test hypothesis regarding the presence of neonicotinoids in the tested water samples was not verified positively, which could result from a number of external factors, notably the occurrence of frequent rainfall in spring and the nearby presence of local rivers which could affect the dilution of the products containing neonicotinoids below the detection levels.
Based on the literature query we assumed that neonicotinoids are well soluble in water and have a long period of decay. They can, therefore, be present and active in the water, soil, crop cultivation, follow-up crops, and plants accompanying crops, even long after they were originally used in a particular crop [26].
- Michel, M, Neonicotinoids – systemic insecticides in plant protection. Progress in plant protection 2020, 60(1), 41-48, DOI: 10.14199/ppp-2020-006.

Reviewer 3 Report
The paper has been substantially improved over the previous version. The one key question that is not addressed involves the use or non-use of neonicotinoids in the fields next to the drainage ditches from which the samples were taken. Also, what time of year were they taken? Some data on this would be extremely useful, perhaps essential in interpreting the results of this paper.
Author Response
Response to the Reviewer 3
Thank you again for your suggestions to better improve our manuscript. We made necessary changes according to your remarks:
- We improved information obout pesticides used by adding information in the manuscript:
It was confirmed through the interviews with 15 farmers and 10 fruit growers that next to 25 points where water from drainage ditches was sampled for testing, pesticides had been used in the spring period in the neighboring 12 rapeseed cultivations, 3 cereal crops cultivations, and 10 fruit orchards:
- in rapeseed fields before the period of rapeseed flowering in May,
- in the cereal crops fields and orchards in April, May, and June,
- and in orchards additionally also in the summer.
Basing on the community interview it was found that the farmers recognised that good yields are obtained using crop protection products and that these products are used in accordance with the financial means - the larger the farm, the more products are used. Products containing neonicotinoids were applied as foliar sprays and as seed coating in seed treatment.

Round 3
Reviewer 1 Report
The authors successfully worked on major remarks. Overall the manuscript could still be improved in terms of the factual 25 sampling test sites and the single! results at each station with exact data. Even if the results are below detection limits for single pesticides at each station, this could be stated in a table from a professional point of view. However apart from this issue, in a qualitative way overall the manuscript is well written by now.